# Time Trends in Prevalence and Antimicrobial Resistance of Respiratory Pathogens in a Tertiary Hospital in Rome, Italy: A Retrospective Analysis (2018–2023)

**DOI:** 10.3390/antibiotics14090932

**Published:** 2025-09-15

**Authors:** Fabio Ingravalle, Massimo Maurici, Antonio Vinci, Stefano Di Carlo, Cartesio D’Agostini, Francesca Pica, Marco Ciotti

**Affiliations:** 1Department of Biomedicine and Prevention, University of Rome Tor Vergata, 00133 Rome, Italy; fabio.ingravalle@students.uniroma2.eu (F.I.); maurici@med.uniroma2.it (M.M.); antonio.vinci@students.uniroma2.eu (A.V.); 2Department of Laboratory Medicine, Tor Vergata University Hospital, 00133 Rome, Italy; stefano.dicarlo@ptvonline.it; 3Department of Experimental Medicine, University of Rome Tor Vergata, 00133 Rome, Italy; cartesio.dagostini@ptvonline.it; 4Laboratory of Microbiology and Virology, Tor Vergata University Hospital, 00133 Rome, Italy

**Keywords:** antibiotic resistance, pathogens prevalence, molecules, respiratory pathogens, retrospective analysis

## Abstract

Background: The increase in antimicrobial resistance (AMR) is a growing concern for global health. Understanding longitudinal trends in pathogen prevalence and resistance patterns is essential for guiding clinical management and antibiotic stewardship. This retrospective observational study analyzed respiratory microbial isolates collected from 2018 to 2023 in Tor Vergata University Hospital, Rome, Italy. Methods: The data were analyzed through WHOnet 2025 software, and the breakpoint references used are those of EUCAST 2025. The data analyzed included pathogen identification, antibiotic resistance rates, and specimen types. Time-trend analyses were conducted to assess changes in pathogen prevalence and antibiotic resistance rates over time, using the Pearson correlation coefficient and linear regression model. Results: More than 54,000 unique microorganism/drug associations were analyzed, with the majority of them relative to inpatients (over 90%). *A. baumannii* showed persistently high prevalence and drug resistance to multiple antibiotics. Significant upward resistance trends of *K. pneumoniae* to multiple antibiotics were observed. Approximately 20% of clinical isolates were fungi, also including some non-albicans Candida (NAC) species, which exhibit intrinsic resistance to azoles. Other microorganisms displayed variable trends in prevalence and resistance profiles. Conclusions: These findings underscore the dynamism of changing patterns of prevalence of microorganisms and their resistance to antimicrobials. They underscore the importance of continuous microbiological surveillance to optimize empirical therapies and guide infection control policies.

## 1. Introduction

Antimicrobial resistance (AMR) is a growing problem in both developed and developing countries. The reduction in available treatment options is placing vulnerable populations, such as older people, children, and immunocompromised patients, at increased risk [1,2]. The increasing spread of AMR among both pathogenic and opportunistic microorganisms is severely limiting therapeutic options and raising concerns that some infections may soon become untreatable [3,4].

This phenomenon arises from multiple, interrelated factors, including the misuse and overuse of antimicrobials (antibiotics and antifungals) in human medicine, as well as their widespread use in agriculture, intensive livestock farming, and food processing, all of which contribute to their massive release into the environment [2,5,6]. Among the various body sites affected by AMR pathogens, the respiratory tract is one of the most impacted, due both to its accessibility to external microorganisms and to the presence of commensal species that can facilitate the transfer of antibiotic resistance plasmids [7,8]. Emerging evidence suggests that the respiratory tract of older patients with chronic conditions represents an important reservoir of antibiotic-resistant microorganisms and that advanced age itself is an independent risk factor associated with the enhanced expression of antibiotic resistance genes [9].

Within this rapidly evolving context, the integration of international and national guidelines for empirical infection management with locally driven surveillance and interventions is essential. This combined strategy not only assists clinicians in choosing the most appropriate therapeutic options, but also empowers public health professionals to design and apply targeted actions—such as environmental control measures (e.g., sanitization, revision of protocols) and health policy initiatives (e.g., vaccination campaigns, health education)—to address and contain the threat of AMR [5,6,7,10].

In 2017, the European Union launched A European One Health Action Plan against Antimicrobial Resistance (AMR), adopting the “One Health” approach that integrates human, animal, and environmental health. Following these recommendations, the Italian government and the regions jointly approved the National Plan to Combat Antimicrobial Resistance (PNCAR) 2017–2020. Through this plan, Italy defined strategies to address AMR at the local, regional, and national levels, aligning its objectives with those of the WHO and the EU within the One Health framework. The authorities subsequently extended the PNCAR until 2021 and introduced a new version for 2022–2025. Despite these efforts, Italy continues to face significant challenges in implementing the plan effectively. Regional differences in resources and organization, limited surveillance coverage, gaps in training for healthcare professionals, and the complexity of coordinating actions across human, veterinary, and environmental sectors all hinder the full achievement of the PNCAR objectives. Strengthening monitoring systems, ensuring uniform application of guidelines, and fostering intersectoral collaboration, therefore, remain essential priorities [11].

Although regional, national, and international initiatives are in place, individual hospitals remain responsible for closely monitoring the circulation of antibiotic-resistant and non-resistant microorganisms within their setting, where numerous variables can significantly influence local epidemiology [1,5,12].

Therefore, the aim of this study is to analyze the prevalence and resistance rates of respiratory pathogens in our hospital over a defined time period and to identify possible significant temporal variations.

## 2. Results

### 2.1. Patients and Clinical Specimens

For this study, we systematically collected data over a period of approximately five years, analyzing microbiological samples from 6953 patients. As shown in Table 1, the age distribution of patients remained relatively stable throughout the study period, with a consistently higher prevalence of males (66.88–71.86%). Regarding specimen types, bronchoaspirate (BAS) represented the most frequent sample, followed by bronchoalveolar lavage (BAL) fluid, which accounted for up to 34% of all specimens in 2020. Oropharyngeal swabs were less frequently collected and showed a gradual decline over time (χ^2^ test, *p* < 0.001). Among the clinical isolates, Gram-negative bacteria were the most frequently detected in respiratory secretions. The prevalence of Gram-positive bacteria and fungi showed significant fluctuations in 2020 and 2021 (χ^2^ test, *p* < 0.001), as reported in Table 1. A detailed description of patient type (inpatient/outpatient) in relation to specimen type and admission/care department is provided in Table A1 of the Appendix B.

### 2.2. Trends in Microbial Isolates in the Observation Period

Table 2 summarizes the frequencies and temporal trends of the 15 most frequently isolated microorganisms during the study period, which together accounted for over 87.33% of all isolates. The most represented species were *P. aeruginosa* (15.85%), *S. aureus* (15.00%), *K. pneumoniae* (12.70%), *C. albicans* (12.08%), and *A. baumannii* (9.15%). Notably, *C. albicans* showed a significantly increasing trend over time (Pearson’s r = 0.51, β = 0.003, *p* = 0.03, R^2^ = 0.26). A rise was also observed for *C. glabrata* and *C. tropicalis* (R^2^ = 0.12 and 0.11, respectively), although these trends did not reach statistical significance (*p* = 0.14 and *p* = 0.16, respectively). By contrast, most bacterial species exhibited no significant temporal changes (*p* > 0.05, low R^2^ values), suggesting overall stable epidemiological patterns.

### 2.3. Resistance Rates

Appendix A provides a detailed overview of all resistance rates and temporal trends for each antibiotic. In total, more than 54,000 unique microorganism–drug associations were analyzed, over 90% of which referred to hospitalized patients.

Among the microorganism/drug associations with the highest resistance, doripenem showed the most critical levels, with an average resistance rate of 93.3% across three pathogens: *A. baumannii*, *P. aeruginosa*, and *K. aerogenes*. Elevated resistance was also observed against amoxicillin/clavulanic acid, with a mean rate of 75.3% in *E. cloacae*, *E. coli*, *K. aerogenes*, *K. pneumoniae*, *P. mirabilis*, and *S. marcescens*. Ciprofloxacin resistance averaged 74.2% across multiple species (*A. baumannii*, *E. coli*, *K. pneumoniae*, *P. aeruginosa*, *P. mirabilis*), while resistance to piperacillin alone (i.e., without tazobactam) reached a mean of 71.7% in *P. aeruginosa* and *K. pneumoniae*. Levofloxacin resistance averaged 66.3%, with the highest rates reported in *A. baumannii*, *K. pneumoniae*, and *S. marcescens*. Notably, *A. baumannii* exhibited the broadest spectrum of resistance, with extremely high rates to ciprofloxacin (93.9%), meropenem (93.3%), imipenem (91.8%), and tobramycin (89.8%). In contrast, *S. aureus*, the only Gram-positive species included, showed very high resistance to penicillin G (>85%), and resistance above 50% to erythromycin, levofloxacin, and oxacillin.

Regarding low-resistance associations, colistin showed the most favorable profile, with six different microorganisms (*A. baumannii*, *E. cloacae*, *E. coli*, *K. oxytoca*, *K. pneumoniae*, *and P. aeruginosa*) exhibiting a mean resistance rate of only 1.9%. Meropenem also displayed very low resistance, averaging 0.7% in *E. coli*, *K. oxytoca,* and *S. marcescens*. Similarly, *K. aerogenes*, *E. coli,* and *K. oxytoca* showed low resistance rates to amikacin. Against the ceftazidime/avibactam combination, *K. pneumoniae*, *E. cloacae*, and *S. marcescens* demonstrated a mean resistance of 1.1%. Finally, *E. coli*, *K. oxytoca*, and *K. aerogenes* exhibited a mean resistance rate of 4.3% to ertapenem.

Resistance to azoles showed variable patterns. Specifically, *C. albicans* and *C. glabrata* exhibited the highest mean resistance to posaconazole, whereas *C. albicans* and *C. tropicalis* showed the lowest resistance rates to itraconazole and fluconazole (see Appendix A). Although *C. glabrata* was not fully resistant to fluconazole, EUCAST guidelines classify it as “susceptible, increased exposure” (breakpoints: I ≤ 16; R ≥ 32). Finally, all tested fungal isolates were susceptible to echinocandins (anidulafungin and micafungin) and amphotericin B.

Table 3 reports the cumulative antibiogram, expressed as susceptibility (“susceptible” + “susceptible, increased exposure” according to EUCAST 2025), for the main microorganisms and antibiotics tested. Data are presented separately for Gram-negative bacteria (Table 3a), Gram-positive bacteria (Table 3b), and fungi (Table 3c).

### 2.4. Resistance Trends

Analysis of the antimicrobial resistance patterns revealed significant variability across the various microorganism/drug associations. Table 4 and Table 5 report the analyzed associations and illustrate the corresponding resistance trends over time, highlighting both increasing and decreasing patterns. Both tables report for each microorganism/drug association analyzed, the breakpoint used for interpreting resistance, the observed resistance rate with its 95% confidence interval, the Pearson correlation coefficient, the regression β-coefficient and its 95% confidence interval, the associated *p*-value, and the R-squared of the regression model.

Table 4 shows that *K. pneumoniae* strains exhibited an increasing resistance over time for multiple antibiotics. *E. cloacae* exhibited a moderately high and increasing resistance rate over time against ceftriaxone.

Table 5 shows all significant decreasing resistance trends by microorganism/drug associations over time. Notably, resistance to dalbavancin in *S. aureus* showed a marked decline (R^2^ = 0.77), as did resistance to itraconazole in *C. tropicalis* (R^2^ = 0.57). Additionally, *A. baumannii* demonstrated a slight downward trend in resistance to ciprofloxacin, imipenem, meropenem, and tobramycin, although these remained consistently high throughout the study period.

## 3. Discussion

### 3.1. Key Results

Our retrospective observational study provides insight into the prevalence and resistance rates of respiratory pathogens in our hospital setting, while also highlighting temporal variations in resistance patterns. The cumulative antibiogram can represent a valuable tool for clinicians, as it not only identifies which drugs have retained efficacy and which have lost it, but also allows a nuanced interpretation of agents with intermediate susceptibility. In particular, the analysis of time trends may offer an early signal of antibiotics that may be progressively losing effectiveness, thereby informing therapeutic choices and guiding local stewardship interventions.

Overall, among the respiratory pathogens identified, the Gram-negative bacteria were the most frequently detected, although in order of frequency, we found *P. aeruginosa*, *S. aureus*, *K. pneumonia*, *C. albicans,* and *A. baumannii*. This distribution reflects local epidemiological patterns, which, although consistent with the international literature, may not be fully generalizable.

Fungi, both *C. albicans* and non-albicans *Candida* species, resulted as emerging pathogens in the patient population, reaching a prevalence of 18%.

Other important findings include the fact that systematic analysis of laboratory data has been able to identify antibiotics that can be recommended locally as empiric treatment while awaiting susceptibility testing, as in the case of colistin and other drugs.

*P. aeruginosa* is an opportunistic microorganism commonly found in the environment and water sources. It may cause severe infections, particularly in immunocompromised hospitalized patients. It is commonly associated with healthcare-associated pneumonia, particularly in patients receiving mechanical ventilation [13,14]. *P. aeruginosa* is a difficult-to-treat pathogen due to its ability to resist antibiotic treatment through multiple intrinsic and acquired antibiotic-resistance mechanisms, biofilm formation, and its capacity to cause persistent infection [15,16]. A surveillance study carried out in Italy by the Italian National Institute of Health (Istituto Superiore di Sanità, ISS) between 2015 and 2023, on blood and liquor samples, observed a substantial decline of *P. aeruginosa* isolates resistant towards the major classes of antibiotics used for the treatment of invasive infections (ceftazidime, fluoroquinolones, piperacillin/tazobactam, carbapenemens, and aminoglycosides) [17]. However, despite this downward trend, resistance rates to these antibiotic classes remain notably high, with 22.1% for piperacillin/tazobactam, 17.8% for ceftazidime, and 16% for both fluoroquinolones and carbapenems. In our study, *P. aeruginosa* isolated from respiratory samples exhibited even higher resistance rates, particularly to piperacillin/tazobactam (47.2%) and piperacillin alone (44.1%). Frequent use of piperacillin/tazobactam in the treatment of *P. aeruginosa* infections may substantially contribute to the selection and emergence of resistant strains [18,19,20]. Resistance tends to develop mainly in older adults, likely due to previous antibiotic exposure as observed in our population. It is also frequently observed in critically ill patients admitted to the ICU, who often present a weakened immune system and are at higher risk of being infected by resistant strains selected by the extensive use of antibiotics in that setting [18].

*S. aureus* is a ubiquitous Gram-positive microorganism that can colonize the skin and mucous membranes of healthy individuals. However, it also represents a major cause of infections, including pneumonia and sepsis, in both community and healthcare settings. In hospitalized and immunocompromised patients, *S. aureus* is frequently associated with severe lower respiratory tract infections, including ventilator-associated pneumonia [21]. The pathogenicity of *S. aureus* is enhanced by its ability to develop resistance to multiple classes of antibiotics, notably methicillin, which defines Methicillin-Resistant *S. aureus* (MRSA), a globally relevant healthcare threat [22,23]. According to the Italian ISS surveillance data on blood and liquor samples, *S. aureus* has shown a slight decreasing trend in MRSA prevalence among invasive isolates over recent years. Nevertheless, resistance rates remain high for erythromycin (37.2%), clindamycin (34.7%), oxacillin/cefoxitin (26.6%), and levofloxacin (24.6%) [24]. In our study, *S. aureus* isolated from respiratory samples showed resistance rates of 58.2% to erythromycin, 5.8% to clindamycin, 58.4% to oxacillin, 38.5% to cefoxitin, and 58.7% to levofloxacin. Except for clindamycin, these values are higher than those reported by the national ISS surveillance. *S. aureus* susceptibility to other molecules was generally similar to that included in the national ISS surveillance.

Resistance in *S. aureus* strains may underline multiple explanations, like an increased circulation of resistant clones in respiratory settings or potential selection due to prior antibiotic exposure [25,26].

*K. pneumoniae* is a Gram-negative bacterium that belongs to the family of Enterobacteriaceae and is commonly present in the human gastrointestinal tract. It is a leading cause of healthcare-associated infections, including pneumonia, bloodstream infections, and urinary tract infections. *K. Pneumoniae* can become an opportunistic pathogen in hospitalized or immunocompromised patients, particularly in intensive care settings and among patients on mechanical ventilation [27,28]. *K. pneumoniae* is of particular concern due to its ability to acquire and disseminate resistance genes, including those encoding extended-spectrum β-lactamases (ESBLs) and carbapenemases, which severely limit treatment options [28].

According to the Italian ISS surveillances data on blood and liquor samples, *K. pneumonia* showed persistently high resistance rates to major antibiotics, including: penicillins (55.4% to amoxicillin/clavulanic acid and 49.0% to piperacillin/tazobactam), cephalosporins (52.1% to cefotaxime, 10.3% to ceftazidime/avibactam and to 49.5% to cefepime), carbapenems (29.8% to imipenem, 25.4% to meropenem and 28.8% to ertapenem), and fluoroquinolones (49.8% to ciprofloxacin and 53.3% to levofloxacin). *K. pneumonia* showed susceptibility to aminoglycosides (resistance rate of 12.5% to amikacin) [29]. In our study, *K. pneumonia* isolates from respiratory samples showed a resistance rate comparable to that reported by the national ISS surveillance. Specifically, resistance was observed to penicillins (55.4% to amoxicillin/clavulanic acid and 49.0% to piperacillin/tazobactam), cephalosporins (63.4% to cefotaxime, 11.5% to ceftazidime/avibactam, and to 66.8% to cefepime), carbapenems (31.0% to imipenem, 44.2% to meropenem, and 58.8% to ertapenem) and fluoroquinolones (67.6% to ciprofloxacin and 70.3% to levofloxacin). In our cohort, *K. pneumonia* showed a similar susceptibility to aminoglycosides, with a resistance rate of 11.1% to amikacin compared to the national rate. The widespread circulation of carbapenemase-producing strains poses a major public health threat in Italy and other Southern European countries [30,31].

A significant increase in the prevalence of *Fungi* was observed between 2020 and 2021, concurrently with the COVID-19 pandemic. Most of the isolated fungal species belonged to the *Candidae* family, among which *C. albicans* showed a significant increase in prevalence. Risk factors for *Candida* colonization of the respiratory tract include the use of broad-spectrum antibiotics, immune suppression, critical illness, mechanical ventilation, and ICU stay [32]. Several studies suggest that co-colonization with *Candida spp.* may worsen patient outcomes and may promote the selection of multidrug-resistant (MDR) bacteria [33,34]. Despite the observed increase in *Candida* prevalence, resistance to antifungal agents appeared to be generally low or absent. This may be due to the infrequent use of antifungal therapy in non-immunocompromised patients, as the presence of *Candida* in respiratory samples is typically interpreted as colonization rather than infection [35]. As fungal treatment is usually reserved for severely immunocompromised patients, the selective pressure for resistance remains limited. However, our findings may raise concerns about the potential emergence of antifungal resistance, particularly among non-*albicans Candida* (NAC) species, to azole compounds. *C. Glabrata* and *P. kudriavzevii* (formerly *C. krusei*—classified in this study in the “other fungi” group due to the low number of isolates) are intrinsically resistant to fluconazole; for this reason, complete susceptibility to the molecule is never foreseen. Other resistance mechanisms include altered drug transport and enzymatic degradation of the antifungal compound. Environmental exposure to azoles, not only in healthcare settings but also in agriculture and other domains, may have contributed to the accelerated development of resistance in fungal species capable of transitioning from saprophytes to opportunistic pathogens, particularly in vulnerable patients [36,37]. These findings highlight the urgent need to implement routine susceptibility testing for fungal isolates, particularly NAC species, even when initially considered mere contaminants. Given the increasing role of environmental fungi not only as emerging hospital pathogens, but also in ecological and agricultural settings, laboratory protocols may require substantial revision to ensure accurate detection and characterization of emerging fungal resistance. Without the implementation of appropriate antifungal stewardship, we may risk repeating the same trajectory observed with antibiotic resistance in livestock farming, where widespread, uncontrolled antimicrobial use contributed to the rapid emergence and spread of resistant strains, now affecting human health [38].

*A. baumannii* was the fifth pathogen detected in order of frequency in the study. Infection by *A. Baumannii* occurs mainly in healthcare settings and can cause sepsis, pneumonia, and urinary tract infections. Risk factors for *A. baumannii* include advanced age, underlying critical illnesses, length of stay in the hospital, reduced immune competence, burns, and mechanical ventilation [39,40]. In Italy, according to the 2023 National Institute of Health (Istituto Superiore di Sanità, ISS) report, *Acinetobacter* spp. present high levels of resistance towards fluoroquinolones (76.9%), carbapenems (75.8%), and aminoglycosides (74.5%) [41]. Our study confirms the national trend showing an even higher level of resistance towards these three classes of antibiotics: carbapenems above 93%, aminoglycosides above 85% and fluoroquinolones above 96% as reported in Appendix A.

Beyond the distribution of the most frequently isolated microorganisms, some antimicrobial agents retained activity against multiple species. Among these, colistin showed the highest preserved efficacy, with very low resistance rates in *K. aerogenes* (≈0%), *E. coli* (0.4%), *P. aeruginosa* (0.9%), *K. oxytoca* (1.9%), *A. baumannii* (2.6%), *K. pneumoniae* (3.4%), and *E. cloacae* (3.9%). In contrast, *S. marcescens* and *P. mirabilis* exhibited extremely high resistance rates to colistin (96.1% and 98.9%, respectively), likely attributable to genomic mechanisms underlying their intrinsic resistance [42]. Amikacin also demonstrated low resistance rates against *E. cloacae* (≈0%), *K. aerogenes* (1.4%), *K. oxytoca* (1.9%), *S. marcescens* (2.6%), *E. coli* (3.5%), *P. aeruginosa* (7.9%), and *K. pneumoniae* (11.1%). However, it was markedly less effective against *A. baumannii* and *P. mirabilis*, which showed resistance rates of 85.9% and 20.2%, respectively. Carbapenems (ertapenem, imipenem, and meropenem) also proved effective against *E. cloacae*, *E. coli*, *K. aerogenes*, *K. oxytoca*, and *S. marcescens* (detailed resistance values are reported in Appendix A).

The increasing resistance trends and high resistance rates underscore the urgent need for coordinated interventions at both local and national levels. These interventions should include stricter antibiotic management and the implementation or reinforcement of antimicrobial stewardship programs. Epidemiological investigations based on microbiological laboratory data can offer valuable insights regarding the presence of molecules that may represent a therapeutic option at the local level, even in the context of high levels of national resistance rates [43].

### 3.2. Limitations

The main limitations of this study are related to its retrospective design and the availability of data for analysis. As a retrospective observational study based exclusively on laboratory records, it does not allow investigation of causal relationships between potential risk factors and the antimicrobial resistance patterns observed. Consequently, we were only able to assess temporal trends without adjusting for possible confounders such as prior antibiotic exposure or clinical severity.

In addition, data on antibiotic consumption at the hospital or departmental level were not available. As a result, we could not evaluate the relationship between antibiotic use pressure and variations in resistance. Moreover, clinical outcome data and molecular resistance data were not available, which limited our ability to correlate microbiological findings with patient prognosis or to explore the genetic mechanisms underlying resistance. Another limitation is the lack of stratification by hospital unit (e.g., ICU vs. non-ICU), which prevents a more granular understanding of resistance dynamics across different clinical settings.

Selection bias is also a potential concern, as microbiological tests were performed only upon clinical request. To mitigate selection bias, diagnostic tests with varying diagnostic reliability were included. This choice was intended to have a study population more closely aligned with the general population. Although this choice introduces heterogeneity into the sample, it reflects the complexity and variability of routine clinical practice in a tertiary care hospital.

Lastly, the role of the COVID-19 pandemic in shaping resistance trends remains unexplored. Given its profound effects on patient populations, infection control, and antimicrobial use, the pandemic may represent a relevant confounder in interpreting our findings.

### 3.3. Interpretation

This study offers an updated overview of the distribution and resistance profiles of respiratory pathogens isolated in a large tertiary care hospital over a five-year period. The predominance of Gram-negative bacteria, particularly *P. aeruginosa*, *K. pneumoniae*, and *A. baumannii*, is consistent with patterns commonly observed in nosocomial respiratory infections [12]. The detection of multiple antimicrobial resistance traits among these isolates further highlights the clinical importance of ongoing microbiological surveillance.

The increasing prevalence of *Fungi*, particularly *C. Albicans*, during the pandemic years aligns with the existing literature describing fungal overgrowth in critically ill and immunocompromised patients [38]. Although our study generally shows low resistance rates versus antifungal molecules, the emergence of resistant NAC species and environmental fungi remains a growing concern, especially in hospital settings where antifungal susceptibility testing is performed only upon specific clinician requests.

Resistance trends observed for *A. Baumannii* were consistent with national and international data. In our cohort, carbapenems, aminoglycosides, and fluoroquinolones showed alarmingly high resistance levels. These findings suggest the presence of local selective pressures, but given the limitations of the study, it is not possible to say whether it is linked to a high use of antibiotics, to a change in infection control protocols during the studied years, or to the importation of resistant species from other sites or facilities.

In this scenario of a dramatic increase in AMR, it is useful to remember that vaccination may have an important role in reducing the burden of antimicrobial resistance through different mechanisms: by preventing infections and thereby reducing antibiotic use, by lowering viral infections that could lead to secondary bacterial infections, and by lowering the risk of spread of resistant strains. Vaccines can limit the specific context in which infections occur and consequently limit the spread of resistant germs. Several studies proved both in pediatric and elderly populations; infection could lead to an increased risk of hospitalization and risk of circulation of resistant bacteria within the hospital setting [44,45].

### 3.4. Generalisability

While the results are consistent with national and international evidence, they may not be fully generalizable beyond the specific setting of a tertiary care hospital. The samples were obtained from different types of respiratory tracts with varying degrees of diagnostic reliability; this heterogeneity may partly reflect real-world variability in clinical practice, thus limiting comparability with other similar studies.

However, the findings remain highly relevant for similar hospital settings, particularly tertiary care centers facing comparable challenges in antimicrobial resistance and infection control. The inclusion of real-world diagnostic variability also enhances the applicability of results to routine clinical practice, albeit with caution when extrapolating to broader or non-hospital populations.

## 4. Materials and Methods

### 4.1. Study Design

This is a retrospective, observational epidemiological study based on microbiology laboratory results. Laboratory data included our types of diagnostic samples: oropharyngeal swab, sputum, bronchial aspirate (BAS), and bronchoalveolar lavage (BAL). All microorganism identification tests with antibiograms/antimycograms were performed upon medical request, primarily to guide and optimize the empirical therapy already initiated either in the hospital or at the patient’s home.

### 4.2. Setting

The study was conducted in Tor Vergata University Hospital, Rome, Italy. Microbiological samples were collected between 1 July 2018 and 30 March 2023. Most samples (90%) were obtained from inpatients, upon a clinician’s request (hospital physician), while the remaining 10% were collected from outpatients referred to for culture and susceptibility testing by their primary care physicians. All patients underwent testing due to suspected respiratory tract infections, for pathogen identification (culture) and resistance profiling (antibiogram/antimycogram). The research team retrieved anonymized data from the laboratory’s electronic records between January and March 2025.

### 4.3. Sample Selection and Study Size

All microbiological samples that tested positive for at least one pathogenic or commensal microorganism of the respiratory tract (bacterial or fungal) were included in the analysis. Samples that tested negative or had saprophytic/commensal flora were excluded. Samples qualitatively and/or quantitatively unsuitable for laboratory analysis, due to collection, transport, or storage defects, were excluded from the study. Repeated samples showing the same microorganism and the same antimicrobial resistance less than 30 days apart on the same patient were excluded. Samples from 6953 patients met the inclusion criteria. A total of 54,305 unique associations of microorganism/drug were included in the study. No a priori sample size calculation was performed, as this was a retrospective observational study based on existing laboratory data.

### 4.4. Variables

For each sample, the data reported below were extracted:A unique sample ID assigned by the microbiology laboratory;Date of sample collection;Patient’s age at the time of sample collection;Patient’s gender;Hospitalization status and department (if hospitalized), or indication of outpatients’ status;Type of respiratory specimen;Identified microorganisms (genus and species);Antimicrobial (antibiotic or antifungal) tested against each microorganism, with resistance data expressed as Minimum Inhibitory Concentration (MIC).

### 4.5. Data Source/Measurements

Microbiological analyses were carried out using Matrix-Assisted Laser Desorption/Ionization—Time of Flight (MALDI-TOF, Bruker Daltonics, Bremen, Germany), and antimicrobial susceptibility was determined by Vitek 2 (bioMérieux, Inc., Hazelwood, MO, USA), according to EUCAST Expert Rules v 3.2.

The measurement of the resistance or susceptibility of each microorganism to each tested molecule was obtained using the WHOnet software (WHOnet V.25.04.25). The data were imported into WHOnet using the BacLink software, provided by WHO with WHOnet. This program, after appropriate configuration, allows you to evaluate the resistance of the entered data, to perform the cumulative antibiogram, and to perform an analysis of the bacterial or fungal strains based on their phenotype [46].

To assess MIC-based resistance or susceptibility, we used the default software settings with breakpoints defined by the EUCAST 2025 guidelines. We selected the EUCAST 2025 reference breakpoints to ensure a standardized evaluation across all observation years, thereby avoiding potential bias in time-trend analysis due to changes in classification cutoffs. This choice also reflects the assumption that the most recent EUCAST guidelines provide improved diagnostic accuracy compared with earlier versions. Data on susceptibility rates and their interpretation were reported following the indications of an independent group regarding the interpretation of the cumulative antibiogram/antimycogram [47].

### 4.6. Bias and Bias Reduction

To reduce bias, inclusion and exclusion criteria were applied to limit the risk of including repeated tests or errors generated by the software in use. To ensure data accuracy, four authors independently examined the electronic database and corrected any identified entry errors. Using an electronic database (Excel), the authors flagged duplicate test records—those with different IDs but matching patient identifiers—and assessed them manually for inclusion or exclusion. Tests performed on the same patient with identical data (same microorganism and same tested molecules) and with a run time of <30 days were excluded to avoid overestimating the prevalence and rates of resistance. The authors did not perform any data imputation.

To evaluate resistance trends over time, the authors applied both Pearson correlation and univariate linear regression analyses. This approach was chosen due to the limited observation window (approximately 5 years) and the absence of key covariates—such as antimicrobial consumption data—which precluded the use of multivariable modelling. Microorganism–antimicrobial associations with fewer than 60 observations across the 19 quarterly intervals were excluded a priori to reduce the risk of autocorrelation and data sparsity, ensuring a sufficient number of data points per time unit.

This study adheres to the STROBe reporting guidelines for cross-sectional studies.

### 4.7. Statistical Methods

The data were analyzed by the Department of Biomedicine and Prevention of Tor Vergata University of Rome.

Descriptive analyses were performed using R software (v. 4.5.1) and R-Studio (v. 2025.05.0 Build 496). The analysis of frequencies, resistance rates, and their corresponding confidence intervals related to bacteria isolates and antimicrobial resistance was performed using WHOnet software (Ver. 4.6.0).

Resistance trend analysis over time (by quarters) was performed using R software (v. 4.5.1) and R-Studio (v. 2025.05.0 Build 496). The analysis included the calculation of the Pearson correlation coefficient and the application of univariate linear regression models, with estimation of the slope coefficient, its corresponding 95% confidence interval, and the model’s R-squared (R^2^) value. Resistance trend analysis was reported in tables with the following elements:Breakpoint—For each antimicrobial agent, the clinical breakpoint defined by the EUCAST 2025 guidelines was reported. This threshold determines whether a microorganism is classified as susceptible, susceptible with increased exposure, or resistant.Observed resistance rate “R (%)”—Expressed as a percentage, this value represents the proportion of resistant isolates among all those tested for a specific microorganism–antibiotic (or antifungal) combination across the entire study period.Ninety-five percent confidence interval (CI) for the resistance rate “IC95 (R%)”—This interval estimates the range within which the true population resistance rate is expected to lie, with 95% confidence, providing a measure of statistical precision.Pearson correlation coefficient “Pearson Coef.”—This coefficient quantifies the strength and direction of the linear relationship between time (in quarters) and the resistance rate. Positive values indicate increasing trends, while negative values suggest decreasing resistance over time.Regression slope coefficient “β coef.”—Derived from a univariate linear regression model, this coefficient represents the estimated average change in resistance rate per quarter. It provides a quantitative measure of the rate of increase or decrease in resistance over time.Ninety-five percent confidence interval for the regression slope coefficient “IC95 β coef.”—This indicates the statistical uncertainty around the estimated slope. If the interval excludes zero, the trend is considered statistically significant.*p*-value—This assesses the statistical significance of the observed trend. A *p*-value < 0.05 indicates that the resistance trend over time is unlikely to be due to chance alone and was considered significant.R-squared “R^2^”—The coefficient of determination quantifies how much of the variation in resistance rates is explained by time in the linear regression model. Values closer to 1 reflect a stronger explanatory power of the model.

## 5. Conclusions

While the results are not generalizable beyond the specific setting, they highlight the importance of conducting periodic resistance monitoring at the local level to inform empiric treatment guidelines and stewardship programs. Actually, our findings indicate that colistin remains an effective option against the major Gram-negative isolates in this study, with the exception of *P. mirabilis* and *S. marcescens*. Notably, it represents the only therapeutic alternative for *A. baumannii*. Similarly, amikacin showed high efficacy against several Gram-negative bacteria, except for *A. baumannii* and *P. mirabilis*. In addition, the ceftazidime/avibactam combination proved effective against multiple Gram-negative species, with the exception of *P. aeruginosa*.

This study also underscores the need to revisit laboratory protocols, particularly regarding the routine susceptibility testing of fungal isolates. The range of antimicrobial agents available to treat infections is limited, and without proper stewardship of antibiotics and antifungals in clinical practice, we risk facing a future where effective therapeutic options for patients are no longer available.

Overall, the findings call for targeted interventions to address antimicrobial resistance in respiratory infections, including better diagnostic stewardship, rational use of antimicrobials, and preventive strategies such as vaccination to reduce the burden of hospital-acquired infections and colonization by resistant organisms.

## Figures and Tables

**Table 1 antibiotics-14-00932-t001:** Demographic characteristics, specimen type, and type of microorganism isolated per year.

Variables		2018 ^α^	2019	2020	2021	2022	2023 ^α^	Total
Age		64.29 ± 16.89	64.26 ± 16.96	67.4 ± 14.15	64.74 ± 15.26	64.41 ± 15.12	63.44 ± 15.56	64.87 ± 15.59
Gender	Male	71.64% (*n* = 384)	71.1% (*n* = 967)	66.84% (*n* = 752)	67.59% (*n* = 1103)	71.76% (*n* = 1324)	71.65% (*n* = 326)	69.84% (*n* = 4856)
	Female	28.36% (*n* = 152)	28.9% (*n* = 393)	33.16% (*n* = 373)	32.41% (*n* = 529)	28.24% (*n* = 521)	28.35% (*n* = 129)	30.16% (*n* = 2097)
Patient	Inpatient	91.79% (*n* = 492)	90.37% (*n* = 1229)	91.11% (*n* = 1025)	93.26% (*n* = 1522)	90.46% (*n* = 1669)	92.97% (*n* = 423)	91.47% (*n* = 6360)
	Outpatient	8.21% (*n* = 44)	9.63% (*n* = 131)	8.89% (*n* = 100)	6.74% (*n* = 110)	9.54% (*n* = 176)	7.03% (*n* = 32)	8.53% (*n* = 593)
Specimen Type	BAS	58.4% (*n* = 313)	54.34% (*n* = 739)	47.82% (*n* = 538)	54.04% (*n* = 882)	56.37% (*n* = 1040)	57.58% (*n* = 262)	54.28% (*n* = 3774)
	BAL	21.64% (*n* = 116)	21.69% (*n* = 295)	34.22% (*n* = 385)	22.12% (*n* = 361)	20.33% (*n* = 375)	19.56% (*n* = 89)	23.31% (*n* = 1621)
	Oropharyngeal swab	9.33% (*n* = 50)	11.03% (*n* = 150)	7.29% (*n* = 82)	7.23% (*n* = 118)	5.31% (*n* = 98)	7.25% (*n* = 33)	7.64% (*n* = 531)
	Sputum	10.63% (*n* = 57)	12.94% (*n* = 176)	10.67% (*n* = 120)	16.61% (*n* = 271)	17.99% (*n* = 332)	15.6% (*n* = 71)	14.77% (*n* = 1027)
Microorganism	Gram-	59.89% (*n* = 321)	64.71% (*n* = 880)	63.73% (*n* = 717)	61.64% (*n* = 1006)	59.02% (*n* = 1089)	57.8% (*n* = 263)	61.5% (*n* = 4276)
	Gram+	20.71% (*n* = 111)	18.68% (*n* = 254)	19.64% (*n* = 221)	15.99% (*n* = 261)	18.43% (*n* = 340)	21.98% (*n* = 100)	18.51% (*n* = 1287)
	Fungi	19.4% (*n* = 104)	16.62% (*n* = 226)	16.62% (*n* = 187)	22.18% (*n* = 362)	22.22% (*n* = 410)	20.22% (*n* = 92)	19.86% (*n* = 1381)
	Others ^β^	0% (*n* = 0)	0% (*n* = 0)	0% (*n* = 0)	0.18% (*n* = 3)	0.33% (*n* = 6)	0% (*n* = 0)	0.13% (*n* = 9)

^α^: for the year 2018, the months from July to December were selected, and for the year 2023, the months from January to March were selected; ^β^: other microorganisms belonging to the kingdom of bacteria not included in the previous classifications.

**Table 2 antibiotics-14-00932-t002:** Isolated microorganisms and their prevalence over time (quarter).

Microorganism	Group	Number of Isolates	(%)	Pearson Coef. ^γ^	β Coef. ^δ^	IC95 β Coef. ^ε^	*p*-Value	R^2^
*P. aeruginosa*	Gram −	1102	15.85	−0.17	−0.001	−0.004, 0.002	0.49	0.03
*S. aureus*	Gram +	1043	15.00	−0.04	−0.000	−0.004, 0.003	0.87	0.00
*K. pneumoniae*	Gram −	883	12.70	0.17	0.000	−0.001, 0.004	0.49	0.03
*C. albicans*	Fungi	840	12.08	0.51	0.003	0.000, 0.005	0.03	0.26
*A. baumannii*	Gram −	636	9.15	−0.22	0.001	−0.000, 0.001	0.36	0.05
*E. coli*	Gram −	423	6.08	−0.14	−0.000	−0.002, 0.001	0.57	0.02
*C. glabrata*	Fungi	202	2.91	0.35	0.001	−0.000, 0.002	0.14	0.12
*S. maltophilia*	Gram −	169	2.43	0.03	0.000	−0.001, 0.001	0.90	0.00
*E. cloacae*	Gram -	150	2.16	0.09	0.000	−0.001, 0.001	0.70	0.01
*C. tropicalis*	Fungi	130	1.87	0.33	0.001	−0.000, 0.001	0.16	0.11
*P. mirabilis*	Gram −	122	1.75	0.02	0.000	−0.001, 0.001	0.95	0.00
*K. aerogenes*	Gram −	119	1.71	−0.36	−0.000	−0.002, 0.000	0.13	0.13
*S. marcescens*	Gram −	105	1.51	−0.23	−0.000	−0.001, 0.000	0.34	0.05
*K. oxytoca*	Gram −	86	1.24	0.00	0.000	−0.001, 0.001	0.99	0.00
*H. influenzae*	Gram −	62	0.89	0.09	0.000	−0.000, 0.001	0.71	0.01
Others Gram-	Gram −	419	6.03	n.a.	n.a.	n.a.	n.a.	n.a.
Others Gram+	Gram +	244	3.51	n.a.	n.a.	n.a.	n.a.	n.a.
Others NAC	Fungi	47	0.68	n.a.	n.a.	n.a.	n.a.	n.a.
Other fungi (not Candida genus)	Fungi	162	2.33	n.a.	n.a.	n.a.	n.a.	n.a.
Others—no classified	n.a.	9	0.13	n.a.	n.a.	n.a.	n.a.	n.a.
Total	n.a.	6953	100	n.a.	n.a.	n.a.	n.a.	n.a.

^γ^: Pearson coefficient is a statistical index that measures the strength and direction of the linear relationship between two variables (positive value for increase, negative value for decrease); ^δ^: regression coefficient is a statistical index that measures how the dependent variable (number of standardized isolates) changes when the independent variable (quarter—time) varies by one unit; ^ε^: 95% confidence interval of the β regression coefficient (inf, sup).

**Table 3 antibiotics-14-00932-t003:** (**a**) Cumulative antibiogram for Gram- bacteria. (**b**) Cumulative antibiogram for Gram+ bacteria. (**c**) Cumulative antimycogram for fungi.

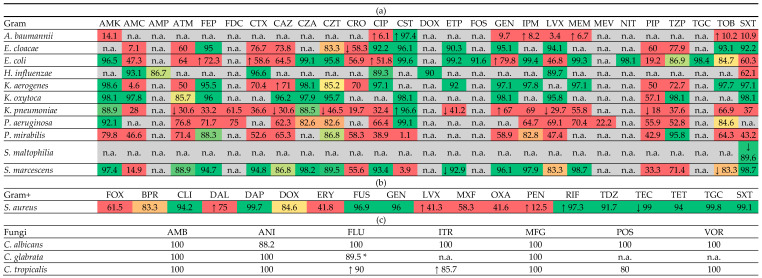

Susceptibility rate >90% (green); 80–90% (from light green to yellow to orange, with different gradations); <80% (red). Arrows indicate temporal trends: ↑ increasing susceptibility during the study period; ↓ decreasing susceptibility during the study period. The drug acronyms are reported in the table of abbreviations at the bottom of the manuscript. * intrinsically resistant to fluconazole; for this reason, complete susceptibility to the molecule is never foreseen.

**Table 4 antibiotics-14-00932-t004:** Microorganism/drug associations with an increasing trend of resistance rates over time.

Organism	Molecules	Break Points		R(%)	IC95 (R%)	Pearson Coef. ^γ^	β Coef. ^δ^	IC95 β Coef. ^ε^	*p*-Value	R^2^
*E. cloacae*	ceftriaxone	S *≤* 1	R *≥* 4	41.7	16.5–71.4	0.827	10.57	0.611; 20.529	0.042	0.685
*K. pneumoniae*	aztreonam	S *≤* 1	R *≥* 8	69.4	56.2–80.1	0.732	3.613	1.598; 5.628	0.002	0.536
*K. pneumoniae*	ceftazidime/ avibactam	S *≤* 8	R *≥* 16	11.5	9.0–14.6	0.601	0.963	0.308; 1.618	0.006	0.361
*K. pneumoniae*	ceftolozane/ tazobactam	S *≤* 2	R *≥* 4	53.5	49.1–57.8	0.594	1.895	0.583; 3.207	0.007	0.353
*K. pneumoniae*	ertapenem	S *≤* 0.5	R *≥* 1	58.8	53.5–64.0	0.61	2.072	0.695; 3.449	0.006	0.372
*K. pneumoniae*	levofloxacin	S *≤* 0.5	R *≥* 2	70.3	65.0–75.1	0.568	1.641	0.424; 2.858	0.011	0.323
*K. pneumoniae*	piperacillin	S *≤* 8	R *≥* 16	82	69.6–90.2	0.742	1.988	0.913; 3.064	0.002	0.551
*S. aureus*	teicoplanin	S *≤* 2	R *≥* 4	1	0.5–2.2	0.548	0.149	0.032; 0.266	0.015	0.3
*S. maltophilia*	trimethoprim/ sulfamethoxazole	I *≤* 2	R *≥* 4	10.4	5.4–18.7	0.533	1.083	0.203; 1.963	0.019	0.284
*S. marcescens*	ertapenem	S *≤* 0.5	R *≥* 1	7.1	1.2–25.0	0.621	3.885	0.631; 7.138	0.023	0.386
*S. marcescens*	tobramycin	S *≤* 2	R *≥* 4	16.7	8.0–30.8	0.57	1.567	0.145; 2.989	0.033	0.325

^γ^: Pearson coefficient is a statistical index that measures the strength and direction of the linear relationship between two variables (positive value for increase, negative value for decrease); ^δ^: regression coefficient is a statistical index that measures how the dependent variable (number of standardized isolates) changes when the independent variable (quarter—time) varies by one unit; ^ε^: 95% confidence interval of the β regression coefficient (inf, sup).

**Table 5 antibiotics-14-00932-t005:** Microorganism/drug associations with a decreasing trend in resistance rates over time.

Organism	Molecules	Break Points		R (%)	IC95 (R%)	Pearson Coef. ^γ^	β Coef. ^δ^	IC95 β Coef. ^ε^	*p*-Value	R^2^
*A. baumannii*	ciprofloxacin	I *≤* 1	R *≥* 2	93.9	91.3–95.9	−0.532	−0.863	−1.565; −0.16	0.019	0.283
*A. baumannii*	colistin	S *≤* 2	R *≥* 4	2.6	1.4–4.6	−0.52	−0.59	−1.086; −0.094	0.023	0.27
*A. baumannii*	imipenem	S *≤* 2	R *≥* 8	91.8	88.0–94.5	−0.579	−0.941	−1.619; −0.263	0.009	0.335
*A. baumannii*	meropenem	S *≤* 2	R *≥* 16	93.3	90.5–95.3	−0.496	−0.794	−1.504; −0.083	0.031	0.246
*A. baumannii*	tobramycin	S *≤* 4	R *≥* 8	89.8	85.7–92.9	−0.501	−1.061	−2.07; −0.053	0.04	0.251
*C. tropicalis*	fluconazole	S *≤* 2	R *≥* 8	10	0.5–45.9	−0.693	−4.575	−8.828; −0.323	0.038	0.48
*C. tropicalis*	itraconazole	S *≤* 0.125	R *≥* 0.25	14.3	0.8–58.0	−0.758	−6.154	−12.244; −0.064	0.048	0.574
*E. coli*	cefepime	S *≤* 1	R *≥* 8	27.7	22.6–33.5	−0.462	−1.193	−2.363; −0.022	0.046	0.214
*E. coli*	cefotaxime	S *≤* 1	R *≥* 4	41.4	34.8–48.4	−0.457	−2.002	−3.998; −0.006	0.049	0.209
*E. coli*	ciprofloxacin	S *≤* 0.25	R *≥* 1	48.2	42.2–54.2	−0.531	−1.888	−3.431; −0.346	0.019	0.282
*E. coli*	gentamicin	S *≤* 2	R *≥* 4	20.2	15.8–25.4	−0.627	−1.706	−2.792; −0.621	0.004	0.393
*K. aerogenes*	ceftazidime/ avibactam	S *≤* 8	R *≥* 16	1.9	0.1–11.2	−0.539	−1.296	−2.509; −0.083	0.038	0.291
*K. pneumoniae*	colistin	S *≤* 2	R *≥* 4	3.4	2.2–5.3	−0.575	−0.381	−0.659; −0.103	0.01	0.33
*K. pneumoniae*	gentamicin	S *≤* 2	R *≥* 4	33	29.3–36.9	−0.478	−1.352	−2.623; −0.081	0.038	0.229
*S. aureus*	dalbavancin	S *≤* 0.25	R *≥* 0.5	25	6.7–57.2	−0.879	−13.35	−21.669; −5.03	0.009	0.773
*S. aureus*	levofloxacin	I *≤* 1	R *≥* 2	58.7	54.9–62.5	−0.707	−1.564	−2.365; −0.763	0.001	0.499
*S. aureus*	penicillin G	S *≤* 0.125	R *≥* 0.25	87.5	84.6–89.9	−0.641	−0.872	−1.406; −0.338	0.003	0.411
*S. aureus*	rifampin	S *≤* 0.064	R *≥* 0.12	2.7	1.7–4.3	−0.638	−0.392	−0.634; −0.15	0.003	0.407

^γ^: Pearson coefficient is a statistical index that measures the strength and direction of the linear relationship between two variables (positive value for increase, negative value for decrease); ^δ^: regression coefficient is a statistical index that measures how the dependent variable (number of standardized isolates) changes when the independent variable (quarter—time) varies by one unit; ^ε^: 95% confidence interval of the β regression coefficient (inf, sup).

## Data Availability

The individual-level data supporting the findings of this study are not publicly available due to privacy and data protection regulations under the General Data Protection Regulation (GDPR, EU Regulation 2016/679). However, aggregated and anonymized data used for the analyses are available in the Appendix A accompanying this article.

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
