# Peer review of "Time Trends in Prevalence and Antimicrobial Resistance of Respiratory Pathogens in a Tertiary Hospital in Rome, Italy: A Retrospective Analysis (2018–2023)"

_antibiotics, 2025, doi:10.3390/antibiotics14090932_

Round 1
Reviewer 1 Report
Comments and Suggestions for Authors
The study investigates temporal trends in the prevalence and antimicrobial resistance profiles of respiratory pathogens isolated in a tertiary care hospital from 2018–2023, aiming to inform local antimicrobial stewardship and empiric therapy guidelines. The topic is relevant and addresses a significant gap that is continuous, hospital-level surveillance data for respiratory pathogens over multiple years. While AMR trend studies exist, this dataset is valuable because it is localized and includes fungal isolates. However, the use of EUCAST 2025 breakpoints for retrospective data interpretation is methodologically questionable, as it may not reflect clinical decision-making during the years studied. Methodology improvements by reassessing breakpoint choice for e.g. For a retrospective study, resistance should be interpreted using the breakpoints in use during each study year (e.g., EUCAST/CLSI versions contemporaneous with sample collection) to reflect the antimicrobial pressures and treatment context for that period. Include antimicrobial consumption data to correlate usage with resistance trends. It would be interesting to include antimicrobial consumption data to correlate usage with resistance trends. The conclusions are broadly consistent with the data presented, especially regarding high resistance rates in Gram-negative bacteria and the need for continuous surveillance. However, given the retrospective breakpoint application issue, some year-to-year resistance trend interpretations may be biased. Adjusting methodology here would make the conclusions more robust. References are generally appropriate, up-to-date, and cover both local/national AMR surveillance and international AMR concerns. Tables are comprehensive.
Additional methodological comment Although the study used EUCAST 2025 breakpoints uniformly, this does not reflect the breakpoints under which patients were clinically treated in earlier years. For accuracy in representing antimicrobial pressures and guiding historical stewardship interpretation, it is recommended to re-analyze using the breakpoints contemporaneous to each study year. This would better capture resistance dynamics relative to actual clinical practice during the study period.
Comments on the Quality of English LanguageComment 2. Moderate language editing is required to correct grammatical errors.
Author Response
The study investigates temporal trends in the prevalence and antimicrobial resistance profiles of respiratory pathogens isolated in a tertiary care hospital from 2018–2023, aiming to inform local antimicrobial stewardship and empiric therapy guidelines. The topic is relevant and addresses a significant gap that is continuous, hospital-level surveillance data for respiratory pathogens over multiple years. While AMR trend studies exist, this dataset is valuable because it is localized and includes fungal isolates. However, the use of EUCAST 2025 breakpoints for retrospective data interpretation is methodologically questionable, as it may not reflect clinical decision-making during the years studied. Methodology improvements by reassessing breakpoint choice for e.g. For a retrospective study, resistance should be interpreted using the breakpoints in use during each study year (e.g., EUCAST/CLSI versions contemporaneous with sample collection) to reflect the antimicrobial pressures and treatment context for that period. Include antimicrobial consumption data to correlate usage with resistance trends. It would be interesting to include antimicrobial consumption data to correlate usage with resistance trends. The conclusions are broadly consistent with the data presented, especially regarding high resistance rates in Gram-negative bacteria and the need for continuous surveillance. However, given the retrospective breakpoint application issue, some year-to-year resistance trend interpretations may be biased. Adjusting methodology here would make the conclusions more robust. References are generally appropriate, up-to-date, and cover both local/national AMR surveillance and international AMR concerns. Tables are comprehensive.
Additional methodological comment Although the study used EUCAST 2025 breakpoints uniformly, this does not reflect the breakpoints under which patients were clinically treated in earlier years. For accuracy in representing antimicrobial pressures and guiding historical stewardship interpretation, it is recommended to re-analyze using the breakpoints contemporaneous to each study year. This would better capture resistance dynamics relative to actual clinical practice during the study period.
ANSWER TO REVIEWER #1
First of all, thank you very much for your appreciation of our topic and the value of our dataset.
We understand the reason for your correct observation about the application of EUCAST 2025 and below we list the reasons that led us to opt for this decision.
- First, using year-specific breakpoints in a trend analysis could artificially inflate or underestimate resistance patterns. Such bias may occur when breakpoint definitions change (e.g., if resistance is redefined from R ≥ 2 to R ≥ 4, this might misleadingly suggest a reduction in resistance, even in the absence of any real change in mean MIC values). For this reason, we considered it more appropriate to apply the EUCAST 2025 breakpoints consistently across all years. We also assumed that the most recent EUCAST definitions represent an improvement in diagnostic accuracy compared with earlier versions.
- Second, since this is an epidemiological study based solely on laboratory data, we did not have access to clinical information (e.g., antibiotic prescriptions, treatment outcomes) or hospital-level antibiotic consumption data. We nevertheless appreciate your valuable suggestion.
In summary, due to the absence of additional clinical and drug consumption data, it was not possible to assess resistance dynamics in relation to actual clinical practice during the study period (lines 362–369). The rationale for our methodological choice is also detailed in the Materials and Methods section (lines480-487).

Reviewer 2 Report
Comments and Suggestions for Authors
This study is based on a large number of respiratory samples and provides valuable analysis of antimicrobial resistance trends over five years. The laboratory methods are robust, modern, and follow international guidelines. Measuring statistical significances of the rising or decreasing trends is also very helpful. However, the overall presentation of the findings requires significant improvement before publication.
Major Comments
- The title, abstract, and introduction do not mention the study location, which may confuse readers. This should be clarified upfront.
-
The Results section currently reports data on a very large number of pathogens and antibiotics, which makes the findings difficult to follow. While it is important to present the breadth of the data, the focus can be on the most clinically relevant pathogens (e.g., Klebsiella pneumoniae, Acinetobacter baumannii, Pseudomonas aeruginosa, Staphylococcus aureus), as these are the major drivers of antimicrobial resistance in respiratory infections. For the other less common organisms, summary data can be moved to the supplementary material. Over AST data (antibiogram) for different pathogens will be very helpful for the readers. However, this is just a suggestion.
In addition, the current tables are very dense and do not allow the reader to easily grasp resistance trends over time. It would be highly beneficial to include graphical visualizations, such as line charts or heatmaps, to illustrate overall resistance trends for key pathogen–antibiotic combinations. For example, a figure showing how carbapenem resistance in K. pneumoniae has changed over the study period, or a heatmap of resistance rates by year across the major pathogens, would make the results much more accessible.
At present, it is difficult to clearly identify “which antibiotics are losing efficacy” and “which remain effective” from the text and tables alone. Visual summaries would greatly improve readability and ensure that the most important findings stand out to the reader.
- The Discussion would benefit from a short opening summary highlighting the study design and key findings before moving into detailed pathogen-specific comparisons. This will help orient readers and make the narrative more coherent.
- The Conclusion section should be rewritten to align more closely with the study findings. It should highlight the main results and then draw implications for local empiric therapy and stewardship. At present, the Conclusion is too generic and not sufficiently supported by the presented data.
- The Conclusion mentions updating treatment guidelines; however, the Discussion does not provide sufficient pathogen-specific or antibiotic-specific recommendations that could support such updates. I suggest the authors highlight which antimicrobials may no longer be suitable for empiric therapy, and which agents remain effective. This will make the study more actionable for clinicians.
- The Limitations section is well presented but should be expanded. In particular, the lack of antibiotic consumption data, absence of clinical outcome and molecular resistance data, no stratification by hospital unit (e.g., ICU vs non-ICU), and the potential influence of the COVID-19 pandemic on trends should be acknowledged.
Minor Comments
-
The reference list contains repetitions and should be corrected.
-
Some language and formatting edits are needed for clarity and consistency.
Author Response
Reviewer 2
This study is based on a large number of respiratory samples and provides valuable analysis of antimicrobial resistance trends over five years. The laboratory methods are robust, modern, and follow international guidelines. Measuring statistical significances of the rising or decreasing trends is also very helpful. However, the overall presentation of the findings requires significant improvement before publication.
ANSWERS TO REVIEWER# 2
First of all, thank you very much for your positive comments on our study and the methodology used in our investigation.
We agree with you that the overall presentation of our findings requires significant improvements before publication, and therefore we have remodeled both the text and the presentation of data in the tables taking into account all your suggestions.
Major Comments
- The title, abstract, and introduction do not mention the study location, which may confuse readers. This should be clarified upfront.
Yes, we apologize for this omission. We have updated the study location in the title of the revised version of the manuscript.
2. The Results section currently reports data on a very large number of pathogens and antibiotics, which makes the findings difficult to follow. While it is important to present the breadth of the data, the focus can be on the most clinically relevant pathogens (e.g., Klebsiella pneumoniae, Acinetobacter baumannii, Pseudomonas aeruginosa, Staphylococcus aureus), as these are the major drivers of antimicrobial resistance in respiratory infections. For the other less common organisms, summary data can be moved to the supplementary material. Over AST data (antibiogram) for different pathogens will be very helpful for the readers. However, this is just a suggestion.
In addition, the current tables are very dense and do not allow the reader to easily grasp resistance trends over time. It would be highly beneficial to include graphical visualizations, such as line charts or heatmaps, to illustrate overall resistance trends for key pathogen–antibiotic combinations. For example, a figure showing how carbapenem resistance in K. pneumoniae has changed over the study period, or a heatmap of resistance rates by year across the major pathogens, would make the results much more accessible.
At present, it is difficult to clearly identify “which antibiotics are losing efficacy” and “which remain effective” from the text and tables alone. Visual summaries would greatly improve readability and ensure that the most important findings stand out to the reader.
We thank you very much for your comments and suggestions. To improve the readability of the article and its results, we have added a new table (now Table 3) that reports the susceptibility rates (rather than resistance rates) of the pathogens to the tested drugs. In line with the classification recommended by Patricia J. Simner et al. (doi: 10.1128/jcm.02210-21), values >90% are highlighted in green, values between 80–90% in gradient from light green to yellow to orange, with different gradations, and values <80% in red.
We hope that the above changes clarify the data presented and improve reader understanding.
3. The Discussion would benefit from a short opening summary highlighting the study design and key findings before moving into detailed pathogen-specific comparisons. This will help orient readers and make the narrative more coherent.
We thank you for your comment. We have added a short summary of the key findings to improve clarity. Should you have any further suggestions for refinement, we would be pleased to consider them. Lines 205-222.
4. The Conclusion section should be rewritten to align more closely with the study findings. It should highlight the main results and then draw implications for local empiric therapy and stewardship. At present, the Conclusion is too generic and not sufficiently supported by the presented data.
Thank you for your constructive feedback. We have revised the Conclusion section to better align it with the study findings and highlight the key results in relation to their implications for local empiric therapy and stewardship programs, as you suggested. You can find the revised version of the Conclusion in lines 540-557.
5. The Conclusion mentions updating treatment guidelines; however, the Discussion does not provide sufficient pathogen-specific or antibiotic-specific recommendations that could support such updates. I suggest the authors highlight which antimicrobials may no longer be suitable for empiric therapy, and which agents remain effective. This will make the study more actionable for clinicians.
We appreciate your comment. However, we would like to emphasize that the results of this study are not easily generalizable, as even comparable hospitals may differ in both patient populations and pathogen distributions. For this reason, we believe that our findings cannot provide national-level guidance; rather, they are applicable only at the local level, as explained in lines 414-418. Nonetheless, the methodology we employed (resistance rates and time trend analysis) can be adopted by individual facilities to perform both internal comparisons (e.g., between departments) and external comparisons (e.g., with similar hospitals), thereby supporting the identification of potential areas for improvement likewise we asses in tables 3, 4 and 5.
6. The Limitations section is well presented but should be expanded. In particular, the lack of antibiotic consumption data, absence of clinical outcome and molecular resistance data, no stratification by hospital unit (e.g., ICU vs non-ICU), and the potential influence of the COVID-19 pandemic on trends should be acknowledged.
We are thankful for your comment. We have expanded the discussion of the study’s limitations, addressing these points, in lines 362–369 and lines 376–379.
Minor Comments
- The reference list contains repetitions and should be corrected.
We apologize for this. We have eliminated duplicated references.
- Some language and formatting edits are needed for clarity and consistency.
We agree with you, and have amended the text accordingly. In detail, wee revised english language and formatting edit

Reviewer 3 Report
Comments and Suggestions for Authors
The authors analyze the evolution of the antimicrobial susceptibility in respiratory isolates, through approximately five years in one single tertiary hospital center, as a basis for the antimicrobial stewardship program. This is a classical analysis of the Microbiology Service in every hospital, made with a yearly basis, to proportionate updated data for the use of first-line antimicrobials in the empiric therapy in different infections.
This information may be useful for the specific center. However, there are some caveats which must be addressed before knowing the usefulness of the study.
1. Last paragraph of the Introduction: “The aim of this study is to highlight, molecule by molecule, the resistance rates of pathogens colonizing the respiratory tract of patients whose samples were analysed at the laboratory of a tertiary hospital and how resistance trends have changed over time.”
Please, clarify: “colonizing” or “infecting”? Moreover, may the authors explain the reason to analyze the activity of antimicrobials instead of the susceptibility / resistance rates of the main microorganism, as suggested in the Title (respiratory pathogens)?
2. Material and Methods: The four types of diagnostic samples are detailed three times; please, simplify.
3. Material and Methods: May the authors explain the reason to use oropharyngeal swabs as diagnostic sample for respiratory infections? The bacteria or fungi found in this sample are mainly colonizing and not infecting.
4. Material and Methods: “Also, contaminated samples and samples not considered suitable for diagnosis by an expert microbiologist were excluded.” May the authors specify the criteria, to make reproducible the study?
5. Material and Methods: EUCAST 2025 guidelines recommend “Susceptible, increased exposure” instead of “intermediate” (line 472).
6. Material and Methods: Definitions of "Breakpoint”, “Observed resistance rate “R (%)”, “p-value”, among others, are not necessary.
7. Results: In Table 1 most samples are from bronchial aspirate and bronchoalveolar lavage (77.59%) which suggests that patients were in intensive care and/or on mechanical ventilation. May the authors clarify the included population?
8. The Oropharyngeal swabs represent 7.64% of samples. Were them analyzed for clinical indication or for colonization study purpose?
9. Sputum samples are 14.77%, in total 1027 samples. However, there are not Streptococcus pneumoniae isolates. Please, clarify if all the sputum were from hospital-acquired respiratory infections.
10. Results: May the authors express the susceptibility / resistance for microorganisms instead of for antimicrobial activities? In the local or general recommendations for the empiric treatment of respiratory infections, the antimicrobials are chosen depending on the clinical diagnosis and the most frequent etiologies of the specific infection; i.e. community-acquired, health-related or ventilator associated pneumonia. Thus, the information for the potential readers should be more useful if aggregated by specific bacteria and not for activity of any individual antimicrobial (Table 3, 4 and S1).
11. All the manuscript: The full name of bacteria must be used the first time and the first letter of the specie never capitalized: i.e. (line 102) Pseudomonas aeruginosa and not P. Aeruginosa. In the subsequent times: P. aeruginosa and not P. Aeruginosa.
Comments on the Quality of English LanguageThe grammar of the English must be improved through professional help.
Author Response
Reviewer 3
The authors analyze the evolution of the antimicrobial susceptibility in respiratory isolates, through approximately five years in one single tertiary hospital center, as a basis for the antimicrobial stewardship program. This is a classical analysis of the Microbiology Service in every hospital, made with a yearly basis, to proportionate updated data for the use of first-line antimicrobials in the empiric therapy in different infections.
This information may be useful for the specific center. However, there are some caveats which must be addressed before knowing the usefulness of the study.
- Last paragraph of the Introduction: “The aim of this study is to highlight, molecule by molecule, the resistance rates of pathogens colonizing the respiratory tract of patients whose samples were analysed at the laboratory of a tertiary hospital and how resistance trends have changed over time.”
Please, clarify: “colonizing” or “infecting”? Moreover, may the authors explain the reason to analyze the activity of antimicrobials instead of the susceptibility / resistance rates of the main microorganism, as suggested in the Title (respiratory pathogens)?
ANSWER TO REVIEWER #3
Thank you very much for your question. We included only biological samples obtained upon a clinician’s request (hospital physician or primary care physician) following a case of respiratory infection with clinical symptoms, for pathogen identification (culture) and resistance profiling (antibiogram). We clarified it in lines 429-434.
- Material and Methods: The four types of diagnostic samples are detailed three times; please, simplify.
We appreciate your comment. We simplified by eliminating repetitions in the Material and Methods section.
- Material and Methods: May the authors explain the reason to use oropharyngeal swabs as diagnostic sample for respiratory infections? The bacteria or fungi found in this sample are mainly colonizing and not infecting.
Thank you very much for your question. Although this is generally true, the oropharyngeal samples selected for this study came from patients with an upper respiratory tract infection (diagnostic questionnaire required to be completed by the patient's physician making the request to the laboratory). This has been further specified in lines 424-426.
- Material and Methods: “Also, contaminated samples and samples not considered suitable for diagnosis by an expert microbiologist were excluded.” May the authors specify the criteria, to make reproducible the study?
We apologize for the misunderstanding. In this passage, we intended to refer to external contamination resulting from non-compliance with laboratory protocols and the manufacturer’s technical instructions. We have reworded the sentence accordingly in lines 440–441.
- Material and Methods: EUCAST 2025 guidelines recommend “Susceptible, increased exposure” instead of “intermediate” (line 472).
We thank you very much for this observation. We agree with you about the point raised and have amended the text accordingly. Lines 512-513
- Material and Methods: Definitions of "Breakpoint”, “Observed resistance rate “R (%)”, “p-value”, among others, are not necessary.
We thank you for your comment and acknowledge your point. While some of us fully agree with your observation, we believe that not all readers, particularly non-experts, non-researchers, or trainees, share the same level of familiarity with statistical analyses and technical definitions as an expert reviewer. For this reason, we prefer to retain these definitions in the manuscript to ensure clarity for a broader audience. Nonetheless, we remain open to further suggestions for corrections or improvements if you consider them necessary.
- Results: In Table 1 most samples are from bronchial aspirate and bronchoalveolar lavage (77.59%) which suggests that patients were in intensive care and/or on mechanical ventilation. May the authors clarify the included population?
Thank you for your request for clarification. Specifically, 71.43% of the identified samples came from in-patients and 6.16% from out-patients. The samples from the ICU represent 45.55% of the total bronchial aspirate and bronchoalveolar lavage samples.
We have added a small table in appendix A and also included it in this reply, specifying the type of patient (in-patient/out-patient) in relation to the type of specimen and the admission/care department at the time the examination was performed. We hope this will clarify the characteristics of the included population.
- The Oropharyngeal swabs represent 7.64% of samples. Were them analyzed for clinical indication or for colonization study purpose?
They were performed only based on clinical indication. We clarified this point in lines 424-426.
- Sputum samples are 14.77%, in total 1027 samples. However, there are not Streptococcus pneumoniae isolates. Please, clarify if all the sputum were from hospital-acquired respiratory infections.
Thank you for giving us the opportunity to clarify this point. Sputum samples were not only from hospital acquired respiratory infections. Microbiological analysis of sputum samples was performed upon request of the clinician based on clinical evaluation on the single patient. S. pneumoniae was isolated 51 times over the study period. This relatively low frequency may be related to Italy’s policy of mandatory and free vaccination for individuals over 60 years of age and for patients with chronic diseases, regardless of age (PCV20 e PPSV 23).
- Results: May the authors express the susceptibility / resistance for microorganisms instead of for antimicrobial activities? In the local or general recommendations for the empiric treatment of respiratory infections, the antimicrobials are chosen depending on the clinical diagnosis and the most frequent etiologies of the specific infection; i.e. community-acquired, health-related or ventilator associated pneumonia. Thus, the information for the potential readers should be more useful if aggregated by specific bacteria and not for activity of any individual antimicrobial (Table 3, 4 and S1).
We thank you for the opportunity to clarify this critical point of the manuscript. A series of Tables 3a, 3b, and 3c (for gram-negative, gram-positive, and fungi, respectively) to provide a detailed overview of the antibiotics most effective against the pathogens most frequently isolated in this study, have been inserted in the revised version of the manuscript.
- All the manuscript: The full name of bacteria must be used the first time and the first letter of the specie never capitalized: i.e. (line 102) Pseudomonas aeruginosa and notP. Aeruginosa. In the subsequent times: P. aeruginosa and not P. Aeruginosa.
We thank you for this observation. We agree with the point raised and have amended the text accordingly.
The grammar of the English must be improved through professional help.
Thank you for your suggestion. We agree with the point raised and we have revised the text.

Round 2
Reviewer 2 Report
Comments and Suggestions for Authors
The authors have addressed most of my comments adequately. I am happy with the revision.